# The clinicopathologic significance of Tks5 expression of peritoneal mesothelial cells in gastric cancer patients

**Atsushi Sugimoto**[1,2,3], **Tomohisa Okuno**[1,2,3], **Gen Tsujio**[1,2,3], **Tomohiro Sera**[1,2,3], **Yurie Yamamoto**[2,3], **Koji Maruo**[1,2,3], **Shuhei Kushiyama**[1,2,3], **Sadaaki Nishimura**[1,2,3], **Kenji Kuroda**[1,2,3], **Shingo Togano**[1,2,3], **Yuichiro Miki**[1,2,3], **Mami Yoshii**[1], **Tatsuro Tamura**[1], **Takahiro Toyokawa**[1], **Hiroaki Tanaka**[1], **Kazuya Muguruma**[1], **Masaichi Ohira**[1], **Masakazu Yashiro**[1,2,3]*

**1** Department of Gastroenterological Surgery, Osaka City University Graduate School of Medicine, Osaka, Japan, **2** Molecular Oncology and Therapeutics, Osaka City University Graduate School of Medicine, Osaka, Japan, **3** Cancer Center for Translational Research, Osaka City University Graduate School of Medicine, Osaka, Japan

* m9312510@med.osaka-cu.ac.jp

**Data Availability Statement:** All relevant data are within the manuscript and its Supporting Information files.

## Abstract

### Background

Gastric cancer (GC) patients frequently develop peritoneal metastasis. Recently, it has been reported that peritoneal mesothelial cells (PMCs) activated by GC cells acquire a migratory capacity and promote GC cell invasion. The invasiveness of PMCs reportedly depends on the activity of Tks5, an adaptor protein required for invadopodia formation. However, the relationship between clinicopathologic features and Tks5 expression in PMCs has been poorly documented. In this study, we evaluated the clinicopathologic significance of the Tks5 expression of PMCs in GC patients.

### Materials and methods

A total of 110 GC patients who underwent gastrectomy were enrolled in this study. Tks5 expressions in PMCs from the greater omentum, lesser omentum and retroperitoneum were evaluated by immunohistochemistry. We analyzed the correlation between Tks5 expressions in PMCs and the patients' clinicopathologic features.

### Results

Tks5 expression was found in 71 (64.5%) of the 110 patients, while 39 (35.5%) were Tks5-negative. Tks5 positivity was significantly (p = 0.038) associated with a greater tumor depth (i.e., T3/4 compared with T1/T2). Peritoneal recurrence was found in 12 of 98 cases within 3 years of surgery. The 3-year peritoneal recurrence-free survival (PRFS) rate in Tks5-positive cases was significantly poorer than that in Tks5-negative cases (80.1% vs 97.4%, p = 0.024). Multivariate analysis revealed that Tks5 positivity and lymph node metastasis were independent factors for PRFS.

**Funding:** The authors received no specific funding for this work.

**Competing interests:** The authors have declared that no competing interests exist.

## Conclusion

Tks5 is frequently expressed in PMCs in advanced-stage gastric cancer. Tks5 might be a useful predictor for peritoneal recurrence in GC patients.

## Introduction

Peritoneal metastasis frequently develops in gastric cancer (GC) patients, even after curative surgery, and results in poor prognosis [1, 2]. The five-year overall survival rate of GC patients with peritoneal metastasis is only 2% [3], and the overall median survival is 3.1 months [4]. No effective therapy has been discovered [5]. Definitive therapy based on the mechanisms responsible for the peritoneal metastasis has long been desired.

The peritoneal cavity is covered by a superficial monolayer of peritoneal mesothelial cells (PMCs) [6]. It has been reported that the mesothelium covered by mesothelial cells might act as a barrier against the invasion of cancer cells [7]. In the process of peritoneal metastasis, cancer cells initially adhere to the PMCs and then invade the sub-mesothelial layer [8]. However, the role of PMCs in the process of peritoneal metastasis has not been fully understood. PMCs have characteristic features of both epithelial and endothelial cells [9]. PMCs themselves change their morphology during peritoneal metastasis, in a fibroblast-like process called the mesothelial-to-mesenchymal transition [10, 11]. A previous study reported that PMCs might guide cancer cell invasion in peritoneal metastasis [12], and that the invasive activity of cancer cells might be activated by PMCs via Tks5 signaling [13]. Tks5 is an adaptor protein which provides a substrate for Src family tyrosine kinases [14]; it is also known to have pivotal roles in podosome and invadopodia formation [15, 16]. Although Tks5 of PMCs might play an important role for the peritoneal metastatic process in GC, the correlation between Tks5 expression in PMCs and clinicopathologic features has not yet been clarified. Thus, in this study, we evaluated the clinicopathologic significance of Tks5 expression in PMCs of GC patients.

## Materials and methods

### Patients

A total of 110 patients who were histologically confirmed to have primary gastric cancer was enrolled in this study. All patients underwent gastrectomy with regional lymph node dissection at Osaka City University Hospital. Peritoneum, such as greater omentum, lesser omentum and retroperitoneum, were included in surgical specimen. The pathologic stage was determined based on the 8th edition of the Union for International Cancer Control TNM classification of malignant tumors. This study was approved by Osaka City University ethics committee (reference number 924). Informed consent was obtained from all patients.

### Immunohistochemistry of Tks5

The expression of Tks5 in PMCs was evaluated by immunohistochemical study. Briefly, slides were deparaffinized and rehydrated with xylene and graded alcohol series and activated by heating. Endogenous peroxidase was blocked and then sections were incubated with the Tks5 antibody (ProteinTech, Rosemont, MN: 18976-1-AP; 1:200) for 60 minutes at room temperature. Tks5 expression was evaluated by intensity of stained PMCs in greater omentum, lesser omentum and retroperitoneum of surgical specimen respectively. Intensity was given scores 0

−2 (0; no staining, 1+; weak staining, 2+; strong staining). Tks5 expression was categorized into three groups according to the intensity. Tks5 positive was defined as the intensity score 1 + or 2+. To compare Tks5 expression in normal peritoneum, Tks5 expression was examined using 17 cases of healthy peritoneum. We confirmed PMCs by using anti-calretinin antibody (Santa Cruz Biotechnology, Inc.; 1:100), a specific marker for PMCs. The evaluation was performed by two double-blinded distinct observers without being aware of clinical data and outcome.

### Statistical analysis

The association between Tks5 expression of PMCs and clinicopathological variables were assessed by Mann-Whitney U test for continuous variables, and chi-squared test or Fisher's exact test for categorical variables. Recurrence free survival (RFS) and peritoneal recurrence-free-survival (PRFS) were measured from the date of surgery to the date of last follow-up or death. Survival rates were calculated by the Kaplan-Meier method, and survival curves were compared with the log-rank test. Uni- and multivariate analyses for PRFS were conducted with Cox proportional hazards models. Multivariate analysis was performed using variables univariate analyses showing $p$-value < 0.1. Hazard ratios (HRs) and 95% confidence intervals (CIs) were calculated. A $p$-value < 0.05 was considered as statistically significant. All data analysis was conducted by JMP® 13 (SAS Institute Inc., Cary, NC, USA).

## Results

### Immunostaining findings of Tks5

**Fig 1** shows representative immunostaining pictures of Tks5 in PMCs and hematoxylin and eosin (HE) staining of the peritoneum in GC cases. Tks5 was stained in the cytoplasm of PMCs of the greater omentum, lesser omentum and retroperitoneum. Of the total of 110 GC cases, 71 cases (64.5%) were Tks5-positive and 39 cases (35.5%) were Tks5-negative. Tks5 expression was examined using 17 cases of healthy peritoneum. The weak expression of Tks5 was found in the cytoplasm of the monolayer PMCs of the healthy peritoneum (**S1A Fig**). The Calretinin expression was found at PMCs with Tks5 expression the surface of peritoneum of gastric cancer cases (**S1B Fig**). Of the total of 17 healthy cases, 6 cases (35.3%) were Tks5-positive and 11 cases (64.7%) were Tks5-negative. **S1 Table** shows that PMCs in GC cases were significantly more frequently positive for Tks5 than that in healthy cases ($p$ = 0.022).

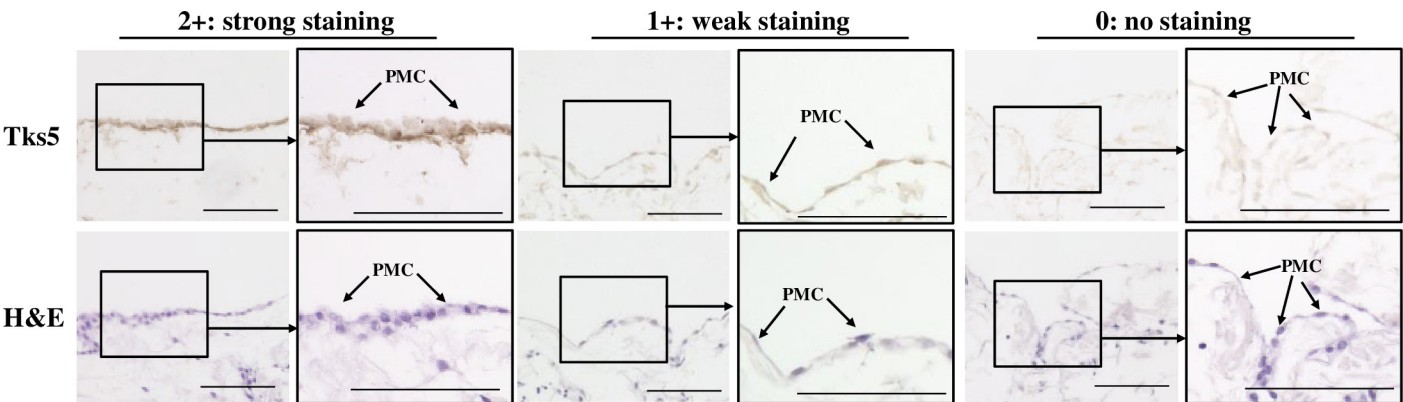

**Fig 1. Representative images of peritoneal mesothelial cells of patients with gastric cancer.** Tks5 expression was found in the cytoplasm of the PMCs forming a monolayer on the greater omentum, lesser omentum, or retroperitoneum in a GC patient. Intensity score: 0; no staining, 1+; weak staining, 2+; strong staining. Positive staining was defined as 1+ or 2+. Arrow, PMC: peritoneal mesothelial cell.

## The relationship between Tks5 expression and clinicopathologic features

The clinicopathologic features of all 110 patients based on the Tks5 expression in PMCs are summarized in **Table 1**. The Tks5-positive group was significantly associated with greater tumor depth (T3/T4) ($p$ = 0.038) compared to the Tks5-negative group, but not other clinico-pathologic features.

## Prognosis

A total of 12 cases of pStage IV were excluded from the PRFS analysis in order to analyze the patients with R0 curative operation. The median follow-up period was 36.7 months (23.3 −46.1 months). Of the total of 98 cases treated with R0 curative operation, recurrence was found in 17 cases (17.3%) within 3 years of the operation, and peritoneal recurrence was 12 (12.2%) of 17 cases (**S2 Table**). **Fig 2** shows overall survival and PRFS based on Tks5 expression in PMCs. The 3-year overall survival showed no significant difference between the Tks5-positive and -negative groups (**Fig 2A**). On the other hand, the 3-year PRFS of the Tks5-positive group was significantly worse than that in Tks5-negative group (Tks5-positive, 80.1%, Tks5-negative, 97.4%, $p$ = 0.024) (**Fig 2B**). In contrast, no significant difference in PRFS or overall survival was found in groups with Tks5-positive and -negative cancer cells (**S2 Fig**).

**Table 1. Relationship between the Tks5 expression of peritoneal mesothelial cells and clinicopathologic features in 110 gastric cancer cases.**

| Variables | | Tks5 expression | | |
|---|---|---|---|---|
| | | Positive (N = 71) | Negative (N = 39) | *p* value |
| Age | years [median (IQR)] | 70 (64 – 78) | 70 (64 – 78) | 0.923 |
| Gender | Female | 29 (40.8%) | 9 (23.1%) | 0.061 |
| | Male | 42 (59.2%) | 30 (76.9%) | |
| Macroscopic type | Borrmann's type 4 | 7 (9.9%) | 2 (5.1%) | 0.387 |
| | Other types | 64 (90.1%) | 37 (94.9%) | |
| Microscopic type | Differentiated | 33 (46.5%) | 24 (61.5%) | 0.129 |
| | Undifferentiated | 38 (53.5%) | 15 (38.5%) | |
| Tumor size | mm [median (IQR)] | 40 (30 – 73) | 40 (20 – 53) | 0.294 |
| Lymphatic invasion | Negative | 41 (57.7%) | 19 (48.7%) | 0.363 |
| | Positive | 30 (42.3%) | 20 (51.3%) | |
| Venous invasion | Negative | 28 (39.4%) | 9 (23.1%) | 0.082 |
| | Positive | 43 (60.6%) | 30 (76.9%) | |
| Depth of invasion | T1/ 2 | 29 (40.9%) | 24 (61.5%) | 0.038 |
| | T3/ 4 | 42 (59.2%) | 15 (38.5%) | |
| Lymph node metastasis | Negative | 31 (43.7%) | 22 (56.4%) | 0.201 |
| | Positive | 40 (56.3%) | 17 (43.6%) | |
| pStage | I and II | 41 (57.7%) | 27 (69.2%) | 0.236 |
| | III and IV | 30 (42.3%) | 12 (30.8%) | |
| Peritoneal cytology | Negative | 65 (91.5%) | 36 (92.3%) | 1* |
| | Positive | 6 (8.5%) | 3 (7.7%) | |
| Peritoneal metastasis | Negative | 65 (91.5%) | 37 (94.9%) | 0.71* |
| | Positive | 6 (8.5%) | 2 (5.1%) | |

* Fisher's exact test.

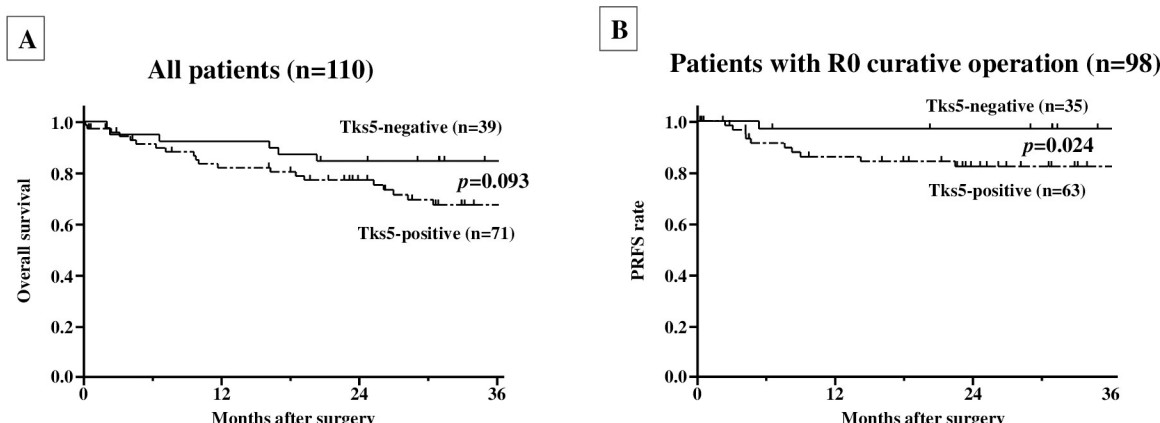

**Fig 2. Survival. A, Overall survival of 110 gastric cancer patients based on Tks5 expression in peritoneal mesothelial cells.** No significant difference was found between the Tks5-positive and -negative groups ($p$ = 0.093). **B, Peritoneal recurrence-free survival of 98 gastric cancer patients based on Tks5 expression in peritoneal mesothelial cells.** The 3-year PRFS of the Tks5-positive group was significantly worse than that in Tks5-negative group ($p$ = 0.024).

### Univariate and multivariate analyses

The results of the univariate and multivariate analyses for PRFS are given in **Table 2**. The univariate analysis showed that poor PRFS was significantly correlated with Tks5 positivity ($p$ = 0.012), female sex ($p$ = 0.026) and lymph node metastasis ($p$ = 0.003). Multivariate analysis revealed that Tks5 positivity ($p$ = 0.039) and lymph node metastasis ($p$ = 0.014) were significantly correlated with poor PRFS.

### Discussion

The present study evaluated the significance of Tks5 expression in PMCs of GC patients. To our best knowledge, this is the first study to correlate Tks5 expression in PMCs with clinico-pathologic features in GC patients. Our data demonstrated that Tks5 expression was

**Table 2. Univariate and multivariate analysis for peritoneal recurrence-free survival.**

| Variables | Univariate analysis | | Multivariate analysis | |
|---|---|---|---|---|
| | HR (95%CI) | $p$ value | HR (95%CI) | $p$ value |
| Tks5 expression | | | | |
| Positive vs Negative | 7.50 (1.45–137) | 0.012 | 5.75 (1.08–106) | 0.039 |
| Age | | | | |
| ≥ 65 vs < 65 years | 2.34 (0.62–15.2) | 0.231 | | |
| Gender | | | | |
| Female vs Male | 3.72 (1.17–13.9) | 0.026 | 2.93 (0.92–11.0) | 0.07 |
| Macroscopic type | | | | |
| Borrmann's type 4 vs Other types | 6.28 (0.95–24.6) | 0.055 | 3.66 (0.54–15.3) | 0.158 |
| Microscopic type | | | | |
| Undifferentiated vs Differentiated | 1.99 (0.63–6.74) | 0.237 | | |
| Tumor size | | | | |
| ≥ 50 vs < 50 mm | 1.75 (0.52–5.49) | 0.351 | | |
| Lymph node metastasis | | | | |
| Positive vs Negative | 7.25 (1.90–47.4) | 0.003 | 5.45 (1.38–36.1) | 0.014 |

Abbreviation: HR, Hazard ratio, CI, Confidence interval.

significantly associated with greater tumor depth (T3/T4). We identified Tks5 expression and lymph node metastasis as independent prognostic factors for PRFS. These findings might indicate that Tks5 expression in PMCs is associated with peritoneal recurrence from GC and that Tks5 could be a predictor of peritoneal recurrence in GC patients with R0 curative surgery.

Tks5 has been found in all cell types including cancer cells and normal cell [17]. The clinical relevance of Tks5 expression in cancer cells has been validated in various cancers, such as prostate cancer [18], breast cancer [16], and melanoma [19], which reported that Tks5 expression of cancer cells is correlated with the progression of carcinomas. However, few reports of Tks5 expression in normal cells have been available in carcinomas. It has been reported that the Tks5 is activated by transforming growth factor-β (TGFβ) via Src phosphorylation [20]. We previously reported that gastric cancer cells increased the Tks5 expression of peritoneal mesothelial cells (PMCs), which stimulated the invasion activity of PMCs and resulted in the stimulation of invasiveness of gastric cancer cells [13]. Since most of gastric cancer cells frequently produce TGFβ [21], TGFβ from gastric cancer cells might activate Tks5 the PMCs via TGFβ-Src-Tks5 signaling. It is conceivable that peritoneal metastasis develops through direct invasion of cancer cells into the gastric wall, exfoliation of free cancer cells from the tumor to the peritoneal cavity, and adhesion to the peritoneum [22, 23]. On the other hand, the clinical relevance of Tks5 in PMCs has not been clarified. In the present study, PMCs in GC cases were significantly more frequently positive for Tks5 than that in healthy cases. These findings indicated that cancer cell might activate Tks5 in PMCs. Tks5 expression in PMCs was significantly associated with tumor depth, which might indicate that deep invasion of GC cells activates Tks5 expression of PMCs. Taken together, Tks5 expression in PMCs was significantly correlated with poor PRFS but not Tks5 expression in cancer cells, indicating that Tks5 expression in PMCs might be a useful predictive marker for peritoneal recurrence in GC patients.

Peritoneal metastasis is the most common metastasis of GC and has a poor prognosis [24]. However, there has been no effective therapy for peritoneal metastasis from GC yet. A previous study reported that peritoneal metastasis from GC was significantly suppressed by knockdown of Tks5 in PMCs [13]. Low expression of Tks5 in tumors was associated with slower tumor growth and less extensive metastasis [25]. Tks5 co-express with ZEB1, which is one of epithelial to mesenchymal transition (EMT) inducers, and Tks5 represent a mediator of invasive behavior of cancer cells [26]. Inhibition of Tks5 resulted in an abrogation of cancer cell extravasation and metastatic colony formation [27]. Tks5 localization is elevated in G1 phase of cell cycle, and anti-proliferative drugs, many of which are arresting cancer cells in G1 phase, may result in higher invasion and metastasis [28]. Tks5 and podosome formation play as mediators of tumor angiogenesis [25]. These findings suggest that Tks5 might be not only a predictive marker but a therapeutic target for prevention of peritoneal recurrence.

In conclusion, Tks5 was found to be frequently expressed in PMCs of advanced-stage gastric cancer patients. PMC Tks5 might be a useful predictor for peritoneal recurrence in surgically treated gastric cancer patients.

## Supporting information

**S1 Fig. Representative images of peritoneal mesothelial cells. A, Representative images of peritoneal mesothelial cells in healthy cases.** The weak expression of Tks5 was found in the cytoplasm of the monolayer PMCs of the healthy peritoneum. Arrow, PMC: peritoneal mesothelial cell. **B, Representative images of peritoneal mesothelial cells in gastric cancer cases.** The Calretinin expression was found at PMCs with Tks5 expression the surface of peritoneum of gastric cancer cases. Arrow, PMC: peritoneal mesothelial cell.
(PPTX)

**S2 Fig. Survival based on Tks5 expression on gastric cancer cells.** No significant difference was found between the Tks5-positive and -negative groups in all patients (A; n = 110) and patients with R0 curative operation (B; n = 98).
(PPTX)

**S1 Table. Tks5 expression of peritoneal mesothelial cells among 110 gastric cancer cases and 17 healthy cases.** Abbreviation: PMC, Peritoneal mesothelial cells.
(DOCX)

**S2 Table. Postoperative recurrence of 98 gastric cancer patients based on Tks5 expression on gastric cancer cells.** * Fisher's exact test. Abbreviation: IQR, Interquartile range.
(DOCX)

**S1 Dataset.**
(XLSX)

## Author Contributions

**Conceptualization:** Atsushi Sugimoto, Masakazu Yashiro.

**Data curation:** Atsushi Sugimoto, Tomohisa Okuno, Gen Tsujio, Tomohiro Sera, Yurie Yamamoto, Koji Maruo, Shuhei Kushiyama, Sadaaki Nishimura, Kenji Kuroda, Shingo Togano, Masakazu Yashiro.

**Formal analysis:** Atsushi Sugimoto, Koji Maruo, Masakazu Yashiro.

**Resources:** Yuichiro Miki, Mami Yoshii, Tatsuro Tamura, Takahiro Toyokawa, Hiroaki Tanaka, Kazuya Muguruma.

**Writing – original draft:** Atsushi Sugimoto, Masakazu Yashiro.

**Writing – review & editing:** Masaichi Ohira.

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
