## [Decision Letter · Decision Letter 0]

18 Mar 2021

PONE-D-21-00313

The Clinicopathologic Significance of Tks5 Expression of Peritoneal Mesothelial Cells in Gastric Cancer Patients

PLOS ONE

Dear Dr. Yashiro,

Thank you for submitting your manuscript to PLOS ONE. After careful consideration, we feel that it has merit but does not fully meet PLOS ONE’s publication criteria as it currently stands. Therefore, we invite you to submit a revised version of the manuscript that addresses the points raised below by the reviewer's during the review process.

We look forward to receiving your revised manuscript.

Kind regards,

Surinder K. Batra

Academic Editor

PLOS ONE

Journal Requirements:

2. Please provide additional details regarding participant consent. In the ethics statement in the Methods and online submission information, please ensure that you have specified what type you obtained (for instance, written or verbal, and if verbal, how it was documented and witnessed).

In addition, please clarify the context in which consent was obtained, and specify whether patients provided:

    1) Consent to use their medical records/samples used in research

    2) Consent to undergo the procedure

    3) Consent to take part in the study reported in this manuscript.

If the ethics committee waived the need for additional informed consent, please state this.

Thank you for your attention to these requests.

3. In the ethics statement in the manuscript and in the online submission form, please provide additional information about the patient records/samples used in your study, including:

a) whether samples were collected prospectively for the purposes of this study, or collected retrospectively from archived samples;

b) whether all data were fully anonymized before you accessed them;

c) the date range (month and year) during which patients' medical records/samples were accessed;

d) the date range (month and year) during which patients whose medical records/samples were selected for this study sought treatment.

4. Please ensure you have discussed any potential limitations of your study in the Discussion, including study design, sample size and/or potential confounders.

'The funders had no role in study design, data collection and analysis, decision to publish, or preparation of the manuscript.'

Please clarify the sources of funding (financial or material support) for your study. List the grants or organizations that supported your study, including funding received from your institution.State what role the funders took in the study. If the funders had no role in your study, please state: “The funders had no role in study design, data collection and analysis, decision to publish, or preparation of the manuscript.”If any authors received a salary from any of your funders, please state which authors and which funders.If you did not receive any funding for this study, please state: “The authors received no specific funing for this work.”

Reviewers' comments:

Reviewer's Responses to Questions

**Comments to the Author**

1. Is the manuscript technically sound, and do the data support the conclusions?

Reviewer #1: Partly

Reviewer #2: Partly

2. Has the statistical analysis been performed appropriately and rigorously? 

Reviewer #1: Yes

Reviewer #2: Yes

3. Have the authors made all data underlying the findings in their manuscript fully available?

Reviewer #1: Yes

Reviewer #2: Yes

4. Is the manuscript presented in an intelligible fashion and written in standard English?

Reviewer #1: Yes

Reviewer #2: Yes

5. Review Comments to the Author

Reviewer #1: Peritoneal dissemination is a frequent metastatic pattern of gastric cancer. Peritoneal mesothelial cells activated by gastric cancer cells play an important role in peritoneal carcinomatosis by providing the invasive front. The current work relies on a previously published report on an adaptor protein, Tks5. Tks5 activity has been shown to facilitate peritoneal mesothelial cell migratory capacity and invasiveness. In the current work, the authors have investigated the clinicopathological correlation of Tks5 in PMC in a gastric cancer patient cohort.

1) General comments:

The overall flow of the paper is satisfactory. However, the authors are encouraged to address the concerns and revise their manuscript before resubmission.

2) Detailed comments:

Abstract:

Abstract is compact and easy to understand

Introduction:

Satisfactory and easy to understand

Materials and methods:

satisfactory

Results:

Authors are encouraged to address the following concerns

1. Provide a better image/improved staining quality for Tks5

2. Functional assay to provide insight on how Tks5 is activated in PMC- whether GC cells induce the expression in PMC? What type of stimulus can activate Tks5 expression in PMC?

3. What is the expression status of Tks5 in healthy peritoneum?

4. Functional data to identify Tks5 dependent gene regulation. How the overall gene landscape is altered in absence of Tks5. Does Tks5 regulate EMT gene signatures or other markers that can affect immune cells in the TME?

5. Does Tks5 play any role in chemoresistance or angiogenesis?

Discussion:

Based on current data, it is easy to understand

Conclusion:

Based on the current data provided, it is easy to understand

References:

good

Images/tables (if any):

Not satisfied with the staining quality. Authors are encouraged to provide a better image/improved staining quality in the revised version.

Language quality:

Easy to understand. Good flow throughout the text. There are a few typographical/grammar errors. Authors are encouraged to correct them in the revised version of the manuscript.

Statistic results (if applicable):

Satisfactory

Reviewer #2: The study is well planned and interesting, however this reviewer has a major concern related to non-specific staining visible in IHC images. The IHC images shown in Figure 1 shows staining of both the mesothelial cells and the underlying stroma. The intensity of the stromal staining is also decreasing as the score for mesothelial cells goes down. Hence, the authors should include some additional control to show the specificity of the staining. Staining with a different antibody might validate the earlier analysis.

6. PLOS authors have the option to publish the peer review history of their article (what does this mean?). If published, this will include your full peer review and any attached files.

Reviewer #1: No

Reviewer #2: No

---

## [Author Response · Author response to Decision Letter 0]

5 May 2021

May 5, 2021

Professor Surinder K. Batra

Academic Editor

PLOS ONE

Dear Professor Surinder K. Batra; 

Re; Manuscript: PONE-D-21-00313

Title: The Clinicopathologic Significance of Tks5 Expression of Peritoneal Mesothelial Cells in Gastric Cancer Patients

Authors: Atsushi Sugimoto, et al. 

We greatly appreciate your invitation for us to revise our article “The Clinicopathologic Significance of Tks5 Expression of Peritoneal Mesothelial Cells in Gastric Cancer Patients”. We would like to thank you for a number of comments and suggestions for improvement in our manuscript. We have carefully considered the referees’ comments and have made point-by-point responses as described below. Also, we highlight all changes in the revised manuscript. This manuscript is not being considered in whole or in part by any other journal. All authors are aware of the content of this manuscript.

We hope you will seriously consider this report for publication in PLOS ONE.

Sincerely,

Masakazu Yashiro, MD

Molecular Oncology and Therapeutics, Osaka City University Graduate School of Medicine, 

1-4-3 Asahi-machi, Abeno-ku, Osaka 545-8585, Japan

TEL; (+81) 6-6645-3838 

FAX; (+81) 6-6646-6450

 

We have responded to the Reviewer #1 comments, as follows.

Reviewer #1

Thank you very much for the careful review of the Reviewer #1. We corrected several points according to the descriptions by the Reviewer #1, as described below. We indicated the changes point by point and highlighted them in the revised paper.

1. Provide a better image/improved staining quality for Tks5.

We improved staining and replaced the picture with the better images of Tks5 staining (Figure 1).

2. Functional assay to provide insight on how Tks5 is activated in PMC- whether GC cells induce the expression in PMC? What type of stimulus can activate Tks5 expression in PMC?

 It has been reported that the Tks5 is activated by transforming growth factor-β (TGFβ) via Src phosphorylation [20]. We previously reported that gastric cancer cells increased the Tks5 expression of peritoneal mesothelial cells (PMCs), which stimulated the invasion activity of PMCs and resulted in the stimulation of invasiveness of gastric cancer cells [13]. Since most of gastric cancer cells frequently produce TGFβ [21], TGFβ from gastric cancer cells might activate Tks5 the PMCs via TGFβ-Src-Tks5 signaling. We added these comments in the discussion. (on page 8 line 15- 21).

3. What is the expression status of Tks5 in healthy peritoneum? 

Tks5 expression was examined using 17 cases of normal peritoneum. The weak expression of Tks5 was found in the cytoplasm of the monolayer PMCs of the normal peritoneum (Supplement Figure 1). Tks5 expression of PMCs was statistically compared between gastric cancer cases and normal cases. Of the total of 17 normal cases, 6 cases (35.3%) were Tks5-positive and 11 cases (64.7 %) were Tks5-negative. Tks5-positive expression of PMCs in GC cases were significantly (p = 0.022) more frequent than that in normal cases (Supplement Table 1). We added these comments in the materials and methods, the results and the discussion. (on page 6 line 10-11; on page 4 line 28-29; on page 6 line 8-15).

4. Functional data to identify Tks5 dependent gene regulation. How the overall gene landscape is altered in absence of Tks5. Does Tks5 regulate EMT gene signatures or other markers that can affect immune cells in the TME?

Sundararajan V et al. reported that Tks5 represented a mediator of invasive behavior of cancer cells through the co-expression with ZEB1 which is one of epithelial to mesenchymal transition (EMT) inducers [26]. We added these comments in the discussion. (on page 9 line 4-6).

5. Does Tks5 play any role in chemoresistance or angiogenesis?

Bayarmagnai B et al. reported that Tks5 localization is elevated in G1 phase of cell cycle, and anti-proliferative drugs, many of which are arresting cancer cells in G1 phase, may result in higher invasion and metastasis. [28]. We added comments in the discussion. (on page 9 line 7-9). Blouw B et al. reported that Tks5 and podosome formation play as mediators of tumor angiogenesis [25]. We added some comments in the discussion. (on page 9 line 9-10). 

We have responded to the Reviewer #2 comments, as follows.

Reviewer #2

Thank you very much for the careful review of the Reviewer #2. We correct several points according to the descriptions by the Reviewer #2, as follows.

1. The study is well planned and interesting, however this reviewer has a major concern related to non-specific staining visible in IHC images. The IHC images shown in Figure 1 shows staining of both the mesothelial cells and the underlying stroma. The intensity of the stromal staining is also decreasing as the score for mesothelial cells goes down. Hence, the authors should include some additional control to show the specificity of the staining. Staining with a different antibody might validate the earlier analysis.

We added additional control to show the specificity of the staining including normal peritoneum and an anti-Calretinin antibody, a specific marker for PMCs. Tks5 expression was examined using 17 cases of normal peritoneum. The weak expression of Tks5 was found in the cytoplasm of the monolayer PMCs of the normal peritoneum (Supplement Figure 1A). The Calretinin expression was found at PMCs with Tks5 expression the surface of peritoneum of gastric cancer cases (Supplement Figure 1B). We replaced the picture with the clearer photo for Tks5 staining in Figure 1. We added some comments in the result. (on page 6 line 8-12).

---

## [Decision Letter · Decision Letter 1]

11 Jun 2021

The Clinicopathologic Significance of Tks5 Expression of Peritoneal Mesothelial Cells in Gastric Cancer Patients

PONE-D-21-00313R1

Dear Dr. Yashiro,

We’re pleased to inform you that your manuscript has been judged scientifically suitable for publication and will be formally accepted for publication once it meets all outstanding technical requirements.

Kind regards,

Surinder K. Batra

Academic Editor

PLOS ONE

Additional Editor Comments (optional):

Reviewers' comments:

Reviewer's Responses to Questions

**Comments to the Author**

1. If the authors have adequately addressed your comments raised in a previous round of review and you feel that this manuscript is now acceptable for publication, you may indicate that here to bypass the “Comments to the Author” section, enter your conflict of interest statement in the “Confidential to Editor” section, and submit your "Accept" recommendation.

Reviewer #1: All comments have been addressed

Reviewer #2: All comments have been addressed

2. Is the manuscript technically sound, and do the data support the conclusions?

Reviewer #1: Yes

Reviewer #2: Yes

3. Has the statistical analysis been performed appropriately and rigorously? 

Reviewer #1: Yes

Reviewer #2: Yes

4. Have the authors made all data underlying the findings in their manuscript fully available?

Reviewer #1: Yes

Reviewer #2: Yes

5. Is the manuscript presented in an intelligible fashion and written in standard English?

Reviewer #1: Yes

Reviewer #2: Yes

6. Review Comments to the Author

Reviewer #1: (No Response)

Reviewer #2: After revision the quality of the images and data has been substantially improved. Addition of new sections in the text has also improved the overall quality.

7. PLOS authors have the option to publish the peer review history of their article (what does this mean?). If published, this will include your full peer review and any attached files.

Reviewer #1: No

Reviewer #2: No

---

## [Editor Report · Acceptance letter]

5 Jul 2021

PONE-D-21-00313R1 

The Clinicopathologic Significance of Tks5 Expression of Peritoneal Mesothelial Cells in Gastric Cancer Patients 

Dear Dr. Yashiro:

I'm pleased to inform you that your manuscript has been deemed suitable for publication in PLOS ONE. Congratulations! Your manuscript is now with our production department. 

Kind regards, 

on behalf of

Prof. Surinder K. Batra 

Academic Editor

PLOS ONE